# Progress and Current Limitations of Materials for Artificial Bile Duct Engineering

**DOI:** 10.3390/ma14237468

**Published:** 2021-12-06

**Authors:** Qiqi Sun, Zefeng Shen, Xiao Liang, Yingxu He, Deling Kong, Adam C. Midgley, Kai Wang

**Affiliations:** 1Key Laboratory of Bioactive Materials for the Ministry of Education, College of Life Sciences, Nankai University, Tianjin 300071, China; sunqiqi971018@163.com (Q.S.); kongdeling@nankai.edu.cn (D.K.); 2Department of General Surgery, Sir Run-Run Shaw Hospital, School of Medicine, Zhejiang University, Hangzhou 310016, China; srrshszf@zju.edu.cn (Z.S.); srrshlx@zju.edu.cn (X.L.); 3School of Computing, National University of Singapore, Singapore 119077, Singapore; yingxu.he1998@gmail.com

**Keywords:** artificial bile duct, tissue engineering, biomimetic materials, tissue regeneration

## Abstract

Bile duct injury (BDI) and bile tract diseases are regarded as prominent challenges in hepatobiliary surgery due to the risk of severe complications. Hepatobiliary, pancreatic, and gastrointestinal surgery can inadvertently cause iatrogenic BDI. The commonly utilized clinical treatment of BDI is biliary-enteric anastomosis. However, removal of the Oddi sphincter, which serves as a valve control over the unidirectional flow of bile to the intestine, can result in complications such as reflux cholangitis, restenosis of the bile duct, and cholangiocarcinoma. Tissue engineering and biomaterials offer alternative approaches for BDI treatment. Reconstruction of mechanically functional and biomimetic structures to replace bile ducts aims to promote the ingrowth of bile duct cells and realize tissue regeneration of bile ducts. Current research on artificial bile ducts has remained within preclinical animal model experiments. As more research shows artificial bile duct replacements achieving effective mechanical and functional prevention of biliary peritonitis caused by bile leakage or obstructive jaundice after bile duct reconstruction, clinical translation of tissue-engineered bile ducts has become a theoretical possibility. This literature review provides a comprehensive collection of published works in relation to three tissue engineering approaches for biomimetic bile duct construction: mechanical support from scaffold materials, cell seeding methods, and the incorporation of biologically active factors to identify the advancements and current limitations of materials and methods for the development of effective artificial bile ducts that promote tissue regeneration.

## 1. Introduction

Bile duct injury (BDI) is regarded as a substantial clinical challenge in hepatobiliary surgery due to its serious complications [1]. Dependent on the cause of injury, BDI can be divided into iatrogenic BDI (approximately 90–95% of cases) or traumatic BDI [2]. Hepatobiliary, pancreatic surgery, and gastrointestinal surgery can result in iatrogenic BDI, wherein the damage caused by cholecystectomy accounts for approximately 80% of all iatrogenic BDI incidence. Laparoscopic cholecystectomy is the leading choice for cholecystectomy due to the low occurrence of postoperative complications. However, a retrospective study involving 10,123 patients highlighted that laparoscopic cholecystectomy may not significantly reduce the incidence of iatrogenic BDI, compared to open cholecystectomy. Only a third of iatrogenic BDI are identifiable during open surgery and thus receive timely, effective repair and reconstruction [3].

The majority of iatrogenic BDI are diagnosable by insensitive methods such as clinical manifestation, laboratory-based testing, and invasive imaging examination [4]. Stenosis after BDI is associated with the excessive proliferation of fibroblasts, collagen deposition, and fibrosis leading to excessive scar tissue hyperplasia [5]. Incidence of stenosis is an important factor associated with the high mortality rate after the operation. If left unchecked, stenosis may lead to severe complications, such as recurrent abdominal pain, obstructive jaundice, and biliary peritonitis [6].

To prevent and treat BDI and post-injury stenosis, current clinical and exploratory research into interventions has revealed limitations and drawbacks. The classic surgical repair and anastomosis techniques for BDI and post-injury stenosis, such as bile duct repair with T-tube drainage, end-to-end bile duct anastomosis with T tube drainage, and Roux-en-Y bile duct jejunostomy, often result in high frequency of postoperative complications and high recurrence rates of stenosis [7,8]. Autologous tissue transplantation repair methods, such as using autologous vascularized gallbladder valve, gastrointestinal valve, umbilical vein valve, etc., have received extensive attention and development [9,10,11]. However, these treatment solutions require further investigation and evaluation due to the “repairing trauma with trauma” concept of donor tissue co-morbidity and the inconclusive data regarding long-term effects.

In contrast to the current treatment methods, the application of tissue engineering and materials to reconstruct bile ducts and promote the proliferation of bile duct cells seeks to regenerate bile duct tissues from artificial constructs. Recent advancements in tissue engineering and bioengineering methods for artificial bile duct generation have revealed potentially effective and promising solutions to address the complex clinical problem of BDI repair. This review collects the latest published research to comprehensively summarize the recent advancements and limitations in materials research for artificial bile duct engineering.

## 2. Materials Choice for Artificial Bile Ducts

Scaffold materials, cell-seeded materials, and incorporation of biologically active factor are regarded as the three major elements of objective-based tissue engineering [12]. Among these, the selection of scaffold materials is often considered the most critical element. At present, several materials have been investigated for use in the tissue engineering of artificial bile ducts. The materials used can be further divided into non-degradable and degradable materials.

### 2.1. Non-Degradable Materials

Non-degradable synthetic materials, such as fluoroelastomers and polytetrafluoroethylene (PTFE), are resistant to bile corrosion and have structurally stable properties [13,14]. Non-degradable synthetic polymers possess mechanical strength and have low immunogenicity but may induce a chronic foreign body response and lack bioactivity to stimulate cell and tissue regeneration [15]. However, their hydrophobicity and non-degradability directly relate to their likelihood to cause immune rejection in long-term implantation resulting from chronic activation of the foreign body response [16]. Furthermore, non-degradable materials inhibit the adhesion and proliferation of bile duct cells, which renders BDI repair and stricture recanalization relatively difficult. Therefore, recent studies have rarely focused on using non-degradable materials for bile duct reconstruction.

### 2.2. Degradable Natural Materials

Naturally derived materials have their own advantage and unavoidable limitations. Natural materials, such as decellularized extracellular matrix (ECM) polysaccharides and proteins, are inherently superior in terms of biocompatibility but are difficult to apply to engineering methodology due to temperature sensitivity, complex molecular structure, and lack of resistance to compressive loads. Undoubtedly, the rapid remodeling and bioabsorption of ECM-derived materials form their major advantage for tissue engineering.

An increasing number of researchers have refocused efforts on the use of degradable materials to coordinate the rate of bile duct material degradation with bile duct cell regeneration, with the goal to achieve the replacement of the artificial bile duct materials with natural and functional bile duct tissue. Among the degradable materials, decellularized tissues (such as decellularized lumen tissue, etc.), naturally derived materials (such as collagen), and other bio-derived materials are commonly utilized materials in recent artificial bile duct engineering research [17,18,19,20]. ECM scaffolds successfully prepared from luminal tissues under the action of specific decellularization technology [21], have immunogenic cell components (plasma membrane, membrane-associated antigens, soluble proteins, etc.) removed from the heterogeneous luminal wall, whilst retaining important ECM components (e.g., collagen, fibronectin, elastin, glycosaminoglycans). Despite allogeneic or xenogeneic origins, decellularized ECM scaffolds carry reduced risk of serious immune rejection problems after implantation, more-so if the scaffold can be generated with autologous origin [22]. Hassanein et al. reported that bile duct epithelial cells may be present in liver-specific ECM, suggesting that the liver-derived ECM scaffolds may be suitable for the construction of artificial bile ducts [23]. Decellularized luminal tissue scaffolds benefit from inherent biodegradability and bioactivity that is borne from functional ECM components, which can guide the infiltration and proliferation of bile duct cells [23]. However, research into the use of decellularized ECM materials has encountered problems such as bile leakage and bile duct stenosis [24], which may be related to heterogenous variability of ECM preparations, loss of quality from decellularization, or prolonged inflammation and poor scaffold remodeling resultant from low cellular infiltration into mature and dense ECM networks [25].

### 2.3. Degradable Synthetic Materials

Biodegradable and bioabsorbable synthetic materials, such as the polyester family of degradable polymers, have tunable compressive strength, controllable degradation rates and improved biocompatibility, which has led to their widespread adoption for biomedical applications [26,27]. Polylactic acid (PLA), polycaprolactone (PCL), polyglycolic acid (PGA), and other degradable polymer composites with excellent biocompatibility have been frequently polymerized to form composite artificial bile ducts. Barralet and colleagues demonstrated the cytocompatibility and suitability of polyester scaffolds for supported bile duct cell growth, observing that human bile duct epithelial cells (BECs) survived for at least 6 months on a self-developed PGA scaffolds modified by PCL [28]. Other research teams went a step further by utilizing multiple polymer components to fabricate artificial bile ducts. The copolymerization of PLA, PCL, and PGA to construct artificial bile ducts resulted in the repair of the bile duct mucosa at the implantation site at 12-weeks post-surgery and the artificial bile duct materials were degraded within 4–6 months post-implantation in the body [16,29,30]. Aikawa et al. developed a tissue-engineered bioabsorbable polymer patch from a 50:50 copolymer of PLA:PCL reinforced with PGA fibers. The researchers speculated that their polymer patch partially degraded and had been expelled via the intestinal tract within 5 postoperative weeks, when implanted into porcine bile duct defects. Even so, their data suggested that bile duct regeneration had proceeded, and bile drainage and patency were maintained. The polymer patch promoted regeneration without incidence of stricture within the 4 months of assessed implantation time. All recipient pigs survived the 4 postoperative months, and the graft site was indistinguishable from the native bile duct tissue, possessing columnar epithelium and accessory glands [16].

Li and colleagues performed research on the development of artificial bile duct scaffolds that facilitate imaging the reparative and degradative processes of bile duct tissue and the artificial bile duct scaffolds, respectively. The researchers constructed tubular, composite, and bionic artificial bile ducts from a 3D-printed PCL mesh coated with gelatin-methacryloyl (GelMA) and loaded with ultrasmall superparamagnetic iron oxide (USPIO) nanoparticles, which enabled visualization of the scaffold using magnetic resonance imaging (MRI) [15]. GelMA hydrogel contains the integrin-binding arginine-glycine-aspartic acid (RGD) motif, reported to convey improved cell adhesion, proliferation, and cell spreading [31]. Li et al. used the GelMA coats to seal the PCL mesh without detriment to the PCL mechanical properties. In vitro assessment of seeded bone marrow-derived mesenchymal stem cell (BMSC) showed continuous BMSC proliferation over 10 days of culture, and approximately 90% scaffold coverage rate and 95% survival rate after 13 days of culture, but mechanisms of supported cell growth was not reported. However, it was noted that the addition of USPIO nanoparticles impaired BMSC proliferation and scaffold coverage, highlighting the cytotoxic limitations of metallic nanoparticles and MRI contrast agents. The work suggested that PCL/GelMA scaffolds would be suitable for in vivo transplantation of the constructed biologically active artificial bile duct [15] (Figure 1).

### 2.4. Nanomaterials

The emergence of nanomaterials fabrication—materials with at least one structural dimension within the nanoscale—has led to the facilitation of the implant of cells within biomimicry scaffolds that resemble the targeted tissue microenvironment. Theoretically, nanomaterials can regulate the adhesion, growth, and functional gene expression of seeded cells. The use of nanomaterials for the preparation of artificial bile ducts is still at the initial stage of research. Moazeni and colleagues published an investigation into the use of nanomaterial scaffolds in constructing artificial bile ducts. The researchers used poly (l-lactide-co-glycolide) (PLGA) nanofiber membranes prepared by electrospinning, which supported the adhesion and growth of L929 fibroblasts, as evidenced by scanning electron microscopy (SEM). However, the in vivo biocompatibility and appropriate biomechanical testing of these nanomaterial artificial bile ducts requires verification [32].

## 3. Cell-Seeded Scaffolds

Culture of cell-seeded scaffolds aims to study whether a scaffold material can recapitulate or stimulate the necessary cellular functions of various cell types required to rebuild tissues and serves as an important assessment index for tissue engineering technologies [33]. Artificial bile ducts containing cholangiocytic cells, hepatic stem cell-derived bile duct cells, and mature bile duct cells have showed improvements in the performance of promoting bile duct reconstruction, compared to cell-free scaffolds [7,34,35,36,37].

### 3.1. Stem Cells

As an alternative to primary bile duct cells, cells seeded onto scaffolds can also be bile duct cells derived from stem and progenitor cells. Many research groups are looking to stem cells with self-renewal and multidirectional differentiation potential, such as mesenchymal stem cells (MSCs) and induced pluripotent stem cells (iPSCs), differentiation of these stem cells into bile duct cells as an important source for cell-seeding experiments. MSCs isolated from bone marrow (BMSCs), amniotic membrane tissue (AMSCs), adipose tissue (ADSCs), and amniotic fluid (AFSCs) exhibit rapid proliferation in vitro and have the potential to differentiate into multiple cell types of the three germ layers. Thus, MSCs have an important academic value in cell transplantation and tissue engineering applications.

Liu et al. sought to develop dual-layer tubular scaffolds with different functions for each layer, to achieve optimal biomechanical performance and a microenvironment with suitable cytocompatibility. The inner layer consisted of BMSC-laden GelMA macro-porous hydrogel, conducive to cell growth, expansion, and nutrient transportation. The outer composite layer of crosslinked GelMA/polyethylene glycol diacrylate (PEGDA) provided mechanical support but with a high degree of deformability and flexibility, similar in elastic modulus to porcine bile ducts. The four-step solution casting method employed the sequential downsizing of internal rods, which resulted in tubular cell-seeded scaffolds (Figure 2). The two layers has strong interfacial bonding accelerated by photopolymerization when exposed to UV light. The dual-layered tubular artificial bile ducts avoided biomechanical problems associated with short-term patency whilst maintaining the viability of seeded BMSCs. Moreover, the researchers demonstrated that the BMSCs differentiated to cholangiocytes when induced by bile acids in vitro [38].

Li et al. directly co-cultured BMSCs with their bile duct materials and assessed proliferation and coverage to reflect material biocompatibility [15]. Lin et al. speculated that BECs differentiated from BMSCs could be utilized to repair BDI but have yet utilize BECs as a novel therapy for biliary leakage [39]. Zong et al. [29] and Miyazawa et al. [30] verified their artificial bile duct materials in animal models. Both groups used co-polymer scaffolds of different composite ratios to construct artificial bile ducts, but the scaffold design differed. Zong and colleagues constructed bi-layered bile duct scaffolds comprised of a compact mechanically appropriate inner PCL layer and a multi-porous, rapidly degrading PLGA outer layer seeded with human MSCs. The rationale was to provide a tissue-engineered scaffold that could support long-term luminal bile flow whilst improving the formation of neo-duct tissue. The data suggested that re-epithelialization of treated porcine bile duct defects was enhanced by MSC-seeded scaffolds, compared to scaffolds with no seeded cells, which led the researchers to conclude that the paracrine activity of seeded MSCs hastened regeneration [29]. Miyazawa and colleagues employed their bioabsorbable, porous, PGA-fiber reinforced copolymer blend of PCL:PLA as tubular scaffolds. An additional step was taken to coat the lumens with collagen prior to seeding of BMSCs, the team termed the artificial bile ducts as bile duct organoid units (BDOU). The BDOU were used to replace a segment of the common bile duct in porcine models, by anastomosis between the native common bile duct and descending duodenum. After 6 months of implantation, no biliary leakage complications were observed, and the neo-bile duct tissue morphologically and functionally resembled the native bile duct. However, there were no significant differences in re-epithelialization rates shown by both BMSC-seeded scaffolds or cell-free scaffolds at 10-weeks post-implantation [30]. In both studies, undifferentiated stem cells were seeded onto the scaffolds and then used for animal bile duct implantation. Whether the disparity between cell-seeding outcomes was dependent on the source of the MSCs, differences in cell-seeding number, seeded scaffold region, culture techniques, or scaffold compositions is unclear. Both research groups arrived at the conclusion that the materials used possessed suitable biocompatibility, but the suggested differences in reparative effects may have ultimately been dependent on scaffold design.

### 3.2. Stem Cell-Derived Bile Duct Cells and 3D Organoids

The concept of iPSCs was proposed by Yamanaka and colleagues. The researchers combined overexpressed four stem cell transcription factors (Myc, Oct3/4, Sox2, Klf4) collectively referred to as Yamanaka factors, in skin fibroblasts. The resulting iPSCs are phenotypically similar to embryonic stem cells (ESCs) in terms of morphology, degree of differentiation, and gene expression [40]. The field of application of iPSCs derived bile duct cells is being actively developed [41]. The derivation of iPSCs from autologous cells can avoid immune rejection after transplant and reduce immunogenicity of cell-seeded scaffolds. Combination of iPSC with 3D cell culture technology has been utilized to derive ductal plate-like cells and to construct bile duct organoids in vitro [42,43,44,45,46]. Organoids are 3D tissue-like structures derived from the specialized culture of pluripotent stem cells, organ progenitor cells, primary cells, or a combination of multiple organ-specific cell types. Organoid formation can be induced by autonomous aggregation under certain culture conditions in vitro and are often constructed using collagen or hydrogel biomaterials, such as Matrigel [47]. On traditionally used 2D tissue culture plastics, adherent cells attach to the substrate surface and exhibit flattened morphologies, which may alter proliferation, migration, and differentiation characteristics of cells, giving rise to cellular and functional differences from cells present within the 3D matrices in the body. The use of 2D culture in combination with coated plates, or hydrogels has attempted to mimic the ECM conditions imposed on cells in tissue microenvironments, but the composition, cell–cell interactions, and cell arrangements still differ from in vivo. The emergence of 3D organoid cultures has gone some way to bridge the gap between the deficiencies of 2D cultures and in vivo cell circumstances. Organoids better simulate the in vivo tissue microenvironment and provide a means to establishing important cell–cell and cell–ECM interaction models.

Extrahepatic cholangiocyte organoids (ECO) have established usefulness in the basic research of the bile duct development, function, and drug testing, but may also offer cell sources for cell-seeded scaffolds in the regeneration of bile ducts. Sampaziotis et al. directly used primary cells to construct bile duct 3D organoid cultures [48]. Subsequently, they populated custom-made densified collagen tubes with the primary cell bile duct organoids and implanted the organoid-seeded scaffolds into mouse models of bile duct replacement to show effective restoration of the bile duct and functional attainment. Du et al. seeded mouse cholangiocytes into the micro-channels (approximate diameter 250 µm) of ‘duct-on-a-chip’ devices. The cells became polarized, aligned, and formed tight cell–cell junctions with monolayer permeability and barrier function equivalent to natural rat biliary epithelium [49]. This work highlighted the importance of achieving the rapid formation of a functional biliary epithelium monolayer to recapitulate barrier function. Whilst the ‘duct-on-a-chip’ was appropriately sized to the lumen of mouse bile ducts, achieving similar in vitro models for upsized bile ducts in larger animals presents a challenge that tissue engineers and materials scientists should consider when designing artificial bile duct scaffold materials for rapid epithelialization with a view for clinical relevance.

## 4. Biologically Active Factor-Functionalized Scaffolds

Cell-free scaffold materials form an important aspect of clinically desirable solutions for tissue repair and regeneration. The regeneration of tissues requires control over cell recruitment/motility, which is often influenced by peptide and protein chemoattractants [50]. Functionalized cell-free scaffolds aim to improve bioactivity of base materials to reflected coordinated tissue repair. Bile duct formation requires coordinated cell–cell contact, resulting in the regulation of cell differentiation and morphogenesis [51]. Biliary proliferation is due to increased expression of interleukin-6 (IL-6). Bile duct growth and proliferation are regulated by bile acids, hormones, and cytokines including epithelial growth factor (EGF), transforming growth factor-β (TGF-β), and IL-6; whereas Foxa1 and Foxa2 were shown to regulate bile duct development in mice. In other areas of tissue engineering, such as vascular tissue engineering [52] and liver tissue engineering [53], it is widely used by researchers to directly add factors that promote tissue regeneration to materials. However, there have been few studies that utilized carrier materials to add biologically active factors for the tissue engineering of bile duct tissues. Almost a decade ago, Li et al. proposed to apply bovine skin-derived collagen patches to the repair of spindle-shaped defects of the extrahepatic bile duct in pig models. The researchers loaded collagen-binding basic fibroblast growth factor (CBD-bFGF) into the collagen patches to convey additional mitogenic effects and to induce the timely remodeling of the collagen patches and regeneration of the bile duct. This resulted in the slow release of bFGF during the process of collagen degradation, which enhanced the repair and regeneration of bile duct tissue without incidence of biliary stricture, biliary sludge, or lithogenesis at 24 weeks post-surgery. Moreover, the collagen patches were completely replaced with functional bile duct tissue [18]. Despite limited studies, there are multiple bioactive factors that have potential to be applied to the field of bile duct tissue engineering.

Many factors act in different ways, in parallel, in series, or in combination. The normal physiological function of the bile duct requires a blood supply and is therefore sensitive to ischemia [54]. Hepatocyte growth factor (HGF) can promote angiogenesis and plays a role in mitosis, motility, anti-apoptosis and morphogenesis [55]. Vascular endothelial growth factor (VEGF) and bFGF are key regulators of physiological angiogenesis [56,57], wound healing, and tissue repair [58]. IL-6 is a multifunctional cytokine that regulates the immune response, the acute phase response, inflammation and haemopoiesis in vivo [59], which is a key regulator of normal bile duct growth and bile duct proliferation [60]. TGF-β is a chemotactic factor for fibroblasts [61]. TGF-β1, -β2, and -β3 have all been shown to induce the expression of bile duct markers during embryogenesis [62].

In addition to adding active factors to materials, there is also the use of active factors in vitro culture and cell induction. Lin et al. induced differentiation from BMSCs to BECs using cytokine regimen treatments, including transforming growth factor-β (TGF-β), fibroblast growth factor (FGF), and epidermal growth factor (EGF) supplementation, and so on. The resulting BECs expressed specific markers such as cytokeratin-7, cytokeratin-19, and glutamyl transferase [39]. De Assuncao et al. devised a novel strategy for gradual differentiation of bile duct cells, which adopted a special method of exposing iPSC-derived hepatic progenitor cells to high doses of sonic hedgehog (Shh), jagged-1, and TGF-β for obtaining bile duct cells [44]. Dianat et al. established stable conditions for promoting the differentiation of iPSCs-derived hepatoblasts into functional cholangiocytic-like cells (CLCs), which used a feeder-free culture system supplemented with growth hormone (GH), EGF, interleukin-6 (IL-6), and sodium taurocholate. The bile duct cell line created by this method expressed bile duct-specific markers. Furthermore, the culture systems were suggested to have a 3D-orientated epithelial/root apical polarity required for the formation of functional bile duct organoids [45]. Chen et al. demonstrated that murine liver organoids could be used to generate genetically stable CLCs in vitro [63]

Subsequently, two studies used different methods to generate specific CLCs. Ogawa and colleagues used 3D co-culture of hepatoblasts and OP9 stromal cells, supplemented by HGF, EGF, and TGF-β. They achieved efficient differentiation of iPSCs to bile duct cells [64]. Sampaziotis and colleagues induced differentiation from human iPSCs to CLCs using a ’five-step method’ that involved the addition of activin A, FGF-2, bone morphogenetic protein-4 (BMP4), and so on. They then induced 3D organoid formation by culturing CLCs with Matrigel and EGF-containing William’s medium [65]. Matsui et al. suggested that despite the preparation of iPSCs-derived CLCs, the directed differentiation and remodeling of duct plate cells derived from iPSCs had yet to be achieved. Their identification of *AQP1* as a specific marker gene of bile duct progenitor cells allowed them to discover that a combination of TGF-β2 and EGF induced the differentiation of iPSC-derived hepatic progenitor cells into *AQP1*^+^ duct plate cells. This result provided a useful reference for studies of bile duct developmental mechanisms and the establishment of congenital bile duct disease models [42] (Figure 3). Furthermore, Sampaziotis et al. cultured primary cells with an innovative mixture of EGF, R-spondin, and Dickkopf-related protein-1 (DKK-1) to generate ECOs. The informative data from the above works could be interpreted to suggest that functionalization of scaffolds with a combination of factors may lead to enhanced functional bile duct tissue regeneration.

## 5. Current Limitations and Reasons for Poor Clinical Translation

### 5.1. Limitations of Scaffold Materials

As early as 1962, extra-hepatobiliary clinicians and scientists proposed the utilized refractory materials, such as silver and plastic tubes, to reconstruct extrahepatic bile ducts in children with congenital biliary atresia [66]. However, there are still no artificial bile duct reconstruction technologies that have received approval for clinical use. The translation of artificial bile ducts still has a long way to go. Miyazawa et al. stipulated that artificial materials could be considered fit for purpose if jaundice caused by bile duct obstruction and biliary peritonitis caused by bile leakage do not occur after bile duct reconstruction [29]. Biliary leakage and stenosis of artificial bile duct materials represent major challenges. In the absence of an abdominal drainage devices, bile leakage and accumulation in the abdominal cavity can cause intraperitoneal septic complications, elevating associated risks of liver failure and operative mortality [67]. Mild biliary tract stenosis may have no obvious symptoms and signs, and moderate to severe biliary tract stenosis may cause progressive and severe liver damage with symptoms of cholangitis, especially when accompanied by hepatic artery thrombosis.

Except for bile leakage and stricture, biocompatibility and biomechanical compliance and resilience of artificial bile duct materials require careful consideration. Modifications of material surfaces through graft modification technology, plasma technology and wet chemical treatment technology can help to improve the overall biocompatibility of materials. In terms of stents, biocompatibility is essential for the survival of the cells adhering to the materials and in the attenuation of the inflammatory response associated with surgical intervention and transplantation [68]. Controlling the inflammatory response is particularly important for the bile duct to reduce risk of fibrotic inflammatory stenosis and the eventually blockage of the lumen [69]. The mechanical strength and biomechanics of the bile duct are also unneglectable important aspects, but there is a lack of studies on the mechanical aspects of bile duct tissues. The pressure of the biliary is 0.98–1.37 kPa, consequently the maximal pressure capacity of the extrahepatic bile duct under normal conditions is only 1.37 KPa [70]. If the mechanical strength of the scaffold material is insufficient, the material easily collapses and causes stenosis. The bile duct is a typically soft tissue possessing viscoelastic behavior. Most biomechanical studies focus on compliance and the elastic modulus of materials [71]. In vitro studies on the bile duct of pigs have shown that the tissue has nonlinear anisotropic mechanical properties [72]. The biomechanics of the bile ducts in 7–10-month-old pigs is equivalent to that of human adults. Li et al. [73] reported an elastic modulus of 1.26 ± 0.07 kPa for the human bile duct. The structure of the bile duct and the small intestine are similar. Both are composed of a layer of smooth muscle and a layer of connective tissue. Therefore, the mechanical properties of the small intestine make it a useful tissue to use as a baseline in studies. In summary, currently employed scaffold materials are promising in their application for the repair of BDI and have been imperative in informing the future development of scaffold materials but have been met by biomechanical and biocompatibility issues that may lead to biliary leakage, stricture, and stenosis. These major overarching limitations form the core difficulties in clinical translation of many tissue-engineered luminal tissues, including artificial bile ducts. The experimental limitations of various biodegradable materials are summarized in Table 1.

### 5.2. Limitations of Cell-Seeding Techniques

In addition to the commonly debated issues of cell-seeding of scaffold materials, such as pre-transplant preparation and quality control, tumorigenicity risk, low cell survival, and rapid immune-mediated clearance of engrafted cells, the field of bile duct engineering was hindered by the lack of available cells. However, recent benefits have emerged in the form of progress in the understanding of biliary physiopathology and the availability of cholangiocyte cell lines or methods to recapitulate bile duct cell fate differentiation of hepatic precursor cells, stem cells, and iPSCs using various mixtures of growth factors known to direct bile duct embryogenesis [44,78]. Inspirational design and innovations in artificial bile duct engineering require an understanding of how cell interactions with the tissue microenvironment can help guide the self-organization of cholangiocytes into functional biliary tubes [79,80].

Minimal assessment of biocompatibility by determining the effects of scaffold materials on cell proliferation and adhesion bears limited applicability to in vivo transplantation scenarios. In the culture of primary bile duct cells, especially under 2D culture conditions, lack of proliferation, loss of function, and loss of differentiation/functional phenotypes were observed after several generations, due to the lack of a distinct microenvironment suitable for bile duct cell growth [81]. Ogawa et al. [64] generated human ESC and iPSC-derived functional bile duct cells in culture, and Sampaziotis et al. [43] developed iPSC-derived CLCs using in vitro culture methods. However, application of iPSCs face shortcomings, such as the potential to lead to the occurrence of teratomas and the generation of cell subsets without full lineage maturity. At the same time, generating iPSCs remains inefficient, time consuming, and the cells are genetically defective in most cases. Stem cell therapies have the ever-present ethical controversy surrounding the possibility of their uncontrolled differentiation following transplantation. Organoids represent an important step towards the construction of bile duct tissues in vitro. Michalopoulos et al. [80] transformed hepatocytes into BECs utilizing organoid culture, but the cell population exhibited incidence of incomplete morphology and function. Bile duct organoids were also constructed from primary extrahepatic bile duct cells cultured in vitro for twenty generations, to form extrahepatic cholangiocyte organoids (ECOs) with characteristics and functions akin to natural bile duct tissues [48]. The researchers successfully utilized these organoids to repair extrahepatic biliary injury in mouse gallbladder incisions by seeding the ECOs onto 1 mm thick PGA scaffolds sheets to patch the defect, and for mouse bile duct replacement using ECO populated densified collagen tubes. This study also demonstrated a relevant autologous cell source that can potentially be obtained from gallbladder tissue, but only under the condition of ensuring cell quality; removal of the pathogenic gallbladder and biopsy sampling in clinical settings may compromise cell integrity or derive phenotypes that are not conducive for use in cell-seeded scaffolds. The further elucidation of isolation methodology and the extent to which the lengthy propagation and culture conditions of these cells in vitro affects their phenotype and functional properties. Additionally, as noted by the researchers in their gallbladder repair model, cells from grafted ECOs contributed to stromal tissue repair, indicating the need for extensive fate-tracking experiments to determine the full extent of tissues that organoid-derived cells have the potential to inhabit and influence post-transplantation. Organoid research and application to tissue engineering and cell-seeded materials, whilst promising, is still a relatively new field of research facing multiple obstacles on the long way to clinical translation. Nevertheless, the extensive knowledge that can be gained from organoid/material combinations will undoubtedly play a prominent role in the future of regenerative medicine.

The 2D culture of cells does not replicate the physiology of the bile duct and rarely supports continual liquid flow. Implementation of 3D culture has faced problems such as difficulty in controlling the uniform dispersion of cells within substrate gels. In addition, the cell-seeding process still faces many problems. The most common way of cell seeding is to simply add cell suspension drops to the scaffold for culture, but this method also has its disadvantages, such as low cell inventory efficiency, which is only about 10–25% of the seeding efficiency, although longer cell seeding and culture time can increase the effective attachment of cells, it may lead to higher incidence of pollution or adverse cell changes [82]. At the same time, homogenous cell distribution within the scaffold can be difficult to achieve [83]. Centrifugal seeding is a dynamic seeding method, which can maintain the vitality of cells to a certain extent, but its disadvantage is that it has a certain impact on cell morphology [84]. In theory, tissue engineering can provide organs for those who cannot accept traditional allograft transplantation, but the problem of immune rejection cannot be ignored. Even decellularized materials cannot completely remove their immunogenicity, and studies have shown that major histocompatibility complex (MHC) remains after decellularization [85]. Autologous cells can be used in the process of recellularization, but it faces the problems of insufficient source of healthy cells and limited culture of bile duct epithelial cells in vitro.

### 5.3. Limitations of Biologically Active Factor-Functionalized Scaffolds

Neovascularization (angiogenesis) is the basis for tumor growth and metastasis [86], VEGF is one of the key regulators of vascularization, which mediates vascular permeability and is concerned with malignant effusion. HGF has previously been reported to act as a hemangiogenic factor, as well as a mitogenic factor for a variety of tumor cells [87]. Therefore, vascularization is important, but it is also one of the conditions for the proliferation of tumor cells. The limitation of bFGF is that it is prone to degeneration under normal physiological conditions, leading to its loss of activity [88]. IL-6 can mediate proinflammatory effects (Inflammation-induced IL-6 functions as a natural brake on macrophages and limits GN), meanwhile cytokine-induced bile duct proliferation is considered to be a cause of fibrosis since overexpression of IL-6 may cause fibrosis [89]. Transforming growth factor-β (TGF-β) and epithelial-to-mesenchymal transformation (EMT) play a central role in the progression of liver fibrosis [90]. Therefore, the biologically active factor is not perfect, and its use is limited. In the process of inducing cells and introducing materials, the selection and concentration of active factors should be well controlled in order to avoid side effects as much as possible.

## 6. Future Research Directions

Undoubtedly, there has been progress made in materials research for artificial bile duct engineering. However, there are still problems to be solved. For instance, current research has focused on reconstructing the bile duct based on simplistic tubular structures. Segmental replacement and reconstruction are useful in the commonly observed cases of iatrogenic BDI repair and stenosis of the extrahepatic bile duct. Biliary tract neoplasia, distal cholangiocarcinoma, or adenocarcinoma of the ampulla of Vater may require excision of the tumor and surrounding bile duct tissue [91,92,93]. Current strategies do not consider the functional restoration of physiological tissue structures such as the Oddi sphincter, in cases where biliary tract resection cannot preserve the papilla of Vater. The Oddi sphincter is a small smooth muscle sphincter located at the junction of the bile duct, pancreatic duct, and duodenum. It functions as a valve to control the flow of bile and pancreatic juices into the duodenum and prevents duodenal fluid reflux. Biliary-enteric anastomosis is the most common treatment of BDI and stenosis in clinical practice. This surgical intervention clears the bile duct but some of these patients remove the Oddi sphincter. Baiguera et al. [94] found that cholangiocarcinoma is a delayed complication of cholangiojejunostomy, secondary to intestinal/duodenal fluid reflux caused by Oddi sphincter dysfunction. Taking this into consideration, the future design and engineering of artificial bile ducts should not be limited to segmental replacements. There are examples of combining 3D-printing with medical imaging techniques, such as using computer tomography (CT) and MRI to reconstruct replicas of the entire bile duct [95], but these also lack the addition of sphincter structures. Promising progress has been made in the research of tissue-engineered heart valves [96], which share some similarities in structural functionality to the neuromuscular Oddi sphincter and may serve as an appropriate source of inspiration to inform the future design of an artificial bile ducts with Oddi sphincter substitutes in the form of bionic valves. Previously, hypoxic pre-conditioning of stem cells seeded onto esophageal acellular matrix scaffolds achieved regeneration of esophageal muscle layers [97]. More recently, Thurner et al. demonstrated that myogenic progenitor cell-derived smooth muscle cells could be engrafted and incorporated into the host pyloric sphincter, providing a potential route to realizing a cell therapeutic approach to smooth muscle regeneration of contractile sphincters [74]. However, restoration of nerve supply continues to be a hurdle in the development of engineered tissues and is likely to hinder the achievement of pre-injury/pre-resection regeneration of engineered bile duct. Moreover, the bile duct epithelium and its regulation of bile secretion and absorption are controlled by the autonomic nervous system. It remains to be seen whether the achievement of sphincter-containing bile duct replacements will soon be realized, and in effect, be in line with normal sphincter function.

Intrahepatic bile duct cells have endodermal origin and are derived from hepatoblasts [51,98]. According to previous reports, signaling pathways involving mediators such as Notch, TGF-β, Wnt, Hippo and FGF play important roles in the formation, repair, and dysfunctions of bile ducts [99,100,101,102]. Taking the duality of the Notch pathway as an example, Notch initiates the differentiation of precursor cells into bile duct cells but is also involved in the repair of bile ducts [103]. Notch-1/2/3/4 are isoforms encoded by the same gene and interact with ligands such as Jagged-1/2 and Delta-like-1/3/4 [104]. The interruption of the Notch pathway during bile duct formation or bile duct cell differentiation can lead to bile duct malformations. For instance, when the interaction between Notch-2 and Jagged-1 is missing, Alagille syndrome will occur, resulting in defects in the biliary duct tree during in the early stages of development [105]. Notch signaling also activates a multitude of transcription factors, which in turn direct hepatocyte differentiation into BECs [106]. Hepatic nuclear factor 6 (HNF6) is one of the important transcription factors. Relevant experiments have confirmed that HNF6 gene-deficient mice show a loss of gallbladder and abnormal development of intrahepatic bile ducts [107]. HNF6 also controls the stratification and migration of hepatoblasts to form bile ducts [108]. Thus, disrupted Notch/HNF6 signaling may lead to inhibition of migration of hepatoblasts and loss of differentiation. Bile duct repair using artificial bile duct materials ultimately aims to maintain physiological biliary drainage whilst providing support and regulation for the regeneration of functional bile duct tissues. Ogawa et al. demonstrated the importance of Notch signaling during cholangiocyte differentiation [64], whereas Sampaziotis et al. indicated the functional characteristics of cholangiocytes (bile acid transfer, alkaline phosphatase activity, gamma-glutamyl-transpeptidase activity, physiological responses to secretin, somatostatin, and vascular endothelial growth factor) [43]. Exploitation of important signaling cues, such as Notch, via biomaterial design [109] could serve to regulate cholangiocyte differentiation and rapid gain of associated functions [110]. These considerations form important aspects to guide the future design of artificial bile ducts.

Ideal artificial bile ducts should ensure the coordination of materials degradation and tissue regeneration, and effectively prevent the biliary complications. The construction of artificial bile ducts based on materials that emulate or induce cell physiological mechanisms involved in bile duct homeostasis and regeneration have broad clinical application prospects. Prolonged activation of key signaling pathways and rapid establishment of cell-cell junction contacts involved in biliary cell layer function, or the specific capture of biliary epithelial cells to achieve bile duct regeneration may be achieved by tuning scaffold functionalization and topological modification. Typically, research on artificial bile ducts has not thoroughly explored the full extent of material choice or scaffold functionalization with biologically active factors, nor the ramifications of these modifications on bile tract cellular and molecular mechanisms. Development of next generation artificial bile duct materials that are based on the regulation of physiological mechanisms may be imperative to bile duct formation and are idealistic goals in achieving engineered bile duct substitutes.

Clinically, the urgency for bioengineered bile ducts justifies the continued advancements and progress in the research field. However, compared with other bioengineered materials, such as tissue-engineered blood vessels [111] and heart valves [95], the research of artificial bile ducts is lagging, as evidenced by the gap in numbers of annually published research articles. The majority of artificial bile duct research has evaluated materials in large animal models, such as pigs and dogs. While this provides a realistic vision towards applicability on the human scale, the lack of in vitro results and exploratory research into bile duct cell biology is juxtaposed to achieving successful engineering of artificial bile ducts. Key goals include catering materials to provide bioactivity and an explicit conduciveness to promoting bile duct cell and tissue development. Furthermore, in-depth biocompatibility and biomechanical evaluation are indispensable aspects beneficial to the design and engineering of plausible artificial bile ducts. Repairing or replacing extrahepatic and common bile ducts with artificial bile duct materials will provide new and simpler treatment options for a range of bile duct-associated injuries and diseases. The continued progress and discoveries within the tissue engineering and material science research fields will help realize circumvention of the current limitations and drive development toward a realistic solution for biliary diseases through tissue engineering.

## Figures and Tables

**Figure 1 materials-14-07468-f001:**
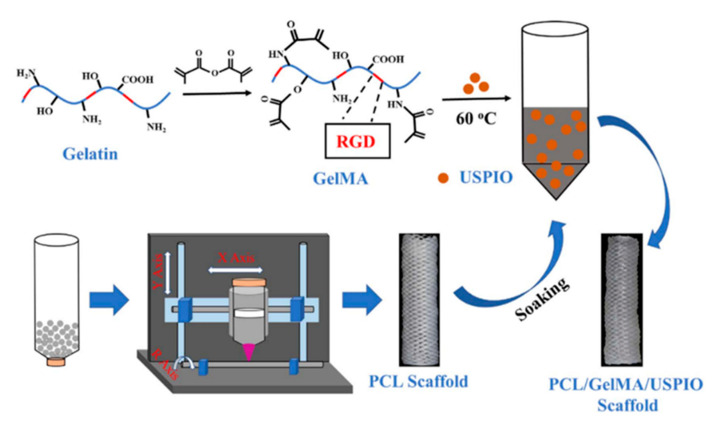
Schematic of a two-step method utilizing 3D-printed tubular polycaprolactone (PCL) scaffolds that were then soaked in a solution of GelMA containing USPIO to produce PCL/GelMA/USPIO composite scaffolds. Figure reproduced with permission [15]. Copyright IOP Publishing.

**Figure 2 materials-14-07468-f002:**
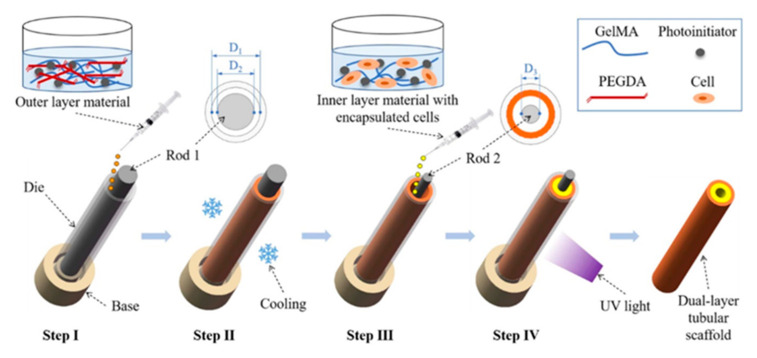
Schematic of a four-step solution casting method for dual-layer tubular scaffold preparation from a cell-encapsulated GelMA inner layer with an outer composite layer of GelMA/PEGDA. Both layers contain photoinitiators that react to UV light to further crosslink the hydrogels and form the final tubular scaffold. Figure reproduced with permission [38]. Copyright IOP Publishing.

**Figure 3 materials-14-07468-f003:**
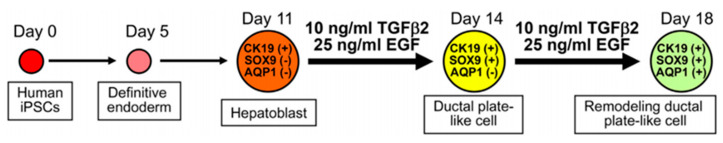
Schematic of stepwise differentiation of human iPSCs into remodeling ductal plate-like cells over the duration of 18 days of in vitro culture. Figure reproduced with permission [42]. Copyright IOP Publishing. Such treatment regimens highlighting a potential route towards the induction of bile duct regeneration using biologically active factor-functionalized scaffolds.

**Table 1 materials-14-07468-t001:** Experimental limitations of materials used for artificial bile duct engineering.

Author, Year, Country	BDI Model	Materials	Limitations
Liu, X. et al., 2021, China [38]	None	BMSC-laden inner GelMA layer and GelMA/PEGDA outer layer	In vitro only, interface adhesion between the two layers is not perfect
Li, H. et al., 2020, China [15]	None	USPIO nanoparticle-loaded and GelMA-coated 3D-printed PCL tubular scaffolds	In vitro only, USPIO inhibited BMSC proliferation, mechanisms of BMSC growth on GelMA not investigated
Park, S.H. et al., 2017, S. Korea [74]	Rabbit common bile duct replacement	3D-printed PVA bile duct replica dip-coated in PCL before PVA removal	Mechanical inadequacies, anastomotic stenosis, mild dilatation of intrahepatic bile duct
Sampaziotis, F. et al., 2017, UK [48]	Mouse extrahepatic BDI	3D cholangiocytic-like cell organoid-seeded collagen tubes	Complex five-step method, unclear impact of long-term in vitro culture on phenotype
Zong, C. et al., 2017, China [29]	Pig extrahepatic BDI	Compact PCL inner layer and multiparous PLGA outer layer	Implants had initial impact on liver function, inflammation, and fibrous hyperplasia after long-term transplant
Struecker, B. et al., 2016, Germany [17]	Pig extrahepatic BDI	Decellularized aorta abdominal aorta, recellularization with bile duct cells	Short experimental duration (14 days), biochemical parameters deviation from normal range, partial implant stenosis
Cheng, Y. et al., 2016, China [24]	Pig extrahepatic BDI	Decellularized ureteral graft	Bile duct stenosis, intrahepatic and extrahepatic bile duct dilatation, stents or T-tubes required
Tao, L. et al., 2014, China [75]	Pig extrahepatic BDI	Freeze-dried collagen membranes	Limited to mesh, fast degradation increased risk of bile leakage and collapse, morphological issues, risk of stenosis.
Pérez Alonso, A.J. et al., 2013, Spain [19]	Guinea pig extrahepatic BDI	Three-dimensional collagen tubes coated with 2% agarose hydrogel	Bile duct marker expression decreased, scaffold not absorbed and granuloma formation at anastomosis site
Li, Q. et al., 2012, China [18]	Pig extrahepatic BDI	CBD-bFGF loading collagen membranes	Limited to mesh, fast degradation increased risk of bile leakage and collapse, morphological issues, risk of stenosis
Nau, P. et al., 2011, USA [76]	Dog extrahepatic BDI	Co-polymer of PGA and trimethylene carbonate	Bile leakage, obstruction, cholangitis
Aikawa, M. et al., 2010, Japan [16]	Pig extrahepatic BDI	Bioabsorbable copolymer patch of PLA/PCL reinforced with PGA fibers	Adhesion, anastomosis deformation, slow cell infiltration
Nakashima, S. et al., 2007, Japan [20]	Dog extrahepatic BDI	Collagen sponge reinforced with polypropylene mesh	Non-biodegradable stents required surgical removal after 2 weeks, prone to stenosis
Barralet, J.E. et al., 2003, UK [28]	None	PGA fiber dip-coated with three layers of PCL	In vitro only
Rosen, M. et al., 2002, USA [77]	Dog extrahepatic BDI	Decellularised small intestine submucosa	Lacked mechanical strength, observed collapse, anastomotic stenosis, and fibrosis, zoonotic risk

## Data Availability

No new data were created or analyzed in this study. Data sharing is not applicable to this article.

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
