# Peer review of "Progress and Current Limitations of Materials for Artificial Bile Duct Engineering"

_materials, 2021, doi:10.3390/ma14237468_

Round 1

Reviewer 1 Report

The paper presents a review of artificial bile duct engineering.  Overall the review presents a topic of interest, but is a bit short and would benefit from further figures highlighting recent work in the field and the exact processes considered. 

Generally, a good length for these reviews that is often recommended for this journal is about 20 pages and 100 references.

Towards the end of the abstract it is written that the review will focus on (1) scaffold materials, (2) cell seeding, and (3) incorporation of biologically active factors.  However, only the first two of these topics receive sections dedicated to their use and limitations.

The only figure provided is Figure 1, it would benefit the paper to have figures demonstrating each of these major processes considered and focused on in the review from the above three points, including more results from other group's work presented as figures.

The writing style and organization could be improved, for instance, the introduction is just one long paragraph, it would benefit from being broken down into multiple paragraphs with more specific focuses.  Likewise the third paragraph of section two is very long.

The first paragraph presents that paper as a systematic review, but no information is provided for how many articles were found for each topic area, were all papers found cited or only a portion of them?

Author Response

The paper presents a review of artificial bile duct engineering. Overall, the review presents a topic of interest, but is a bit short and would benefit from further figures highlighting recent work in the field and the exact processes considered. 

Response: We thank the reviewer for their time and effort in provided constructive criticism and helpful suggestions that would improve our review manuscript. Please find specific responses to the comments below.

Generally, a good length for these reviews that is often recommended for this journal is about 20 pages and 100 references.

Response: After following all reviewers’ suggestions, the article has been expanded. We now provide 111 references and a length just shy of 20 pages.

Towards the end of the abstract it is written that the review will focus on (1) scaffold materials, (2) cell seeding, and (3) incorporation of biologically active factors.  However, only the first two of these topics receive sections dedicated to their use and limitations.

Response: As part of expanding the manuscript, we have added a third section dedicated to the incorporation of biologically factors. We initially included the few studies on this topic within the other sections. However, after careful consideration, we decided to write a much larger third section, which highlights multiple avenues of research that have strong potential to be employed for tissue engineering of bile ducts. We refer the reviewer to sections 4 and 5.3 (limitations of biologically active factor incorporation) in the revised manuscript.

The only figure provided is Figure 1, it would benefit the paper to have figures demonstrating each of these major processes considered and focused on in the review from the above three points, including more results from other group's work presented as figures.

Response: Unfortunately, at the time of original submission deadline, we had not received permissions to include additional figures. In the interim, we received permissions to include additional figures. These are Figure 1 (two-step 3D-printed and functionalized PCL/GelMA/USPIO scaffolds) and Figure 3 (flow diagram of iPSC to ductal-like cell transformation in vitro). Authors of bioactive-factor loaded scaffolds have yet to respond to our requests. We hope the additional figures offer further insight into the described studies.

The writing style and organization could be improved, for instance, the introduction is just one long paragraph, it would benefit from being broken down into multiple paragraphs with more specific focuses.  Likewise the third paragraph of section two is very long.

Response: Duly noted. We broke the introduction down into 5 paragraphs so that each has a specific focal point: BDI and its prevalence, BDI diagnosis, BDI complications, Current/proposed treatment strategies, and the emergence of tissue engineering solutions for BDI. Furthermore, the indicated paragraph in the second section was split into two paragraphs.

The first paragraph presents that paper as a systematic review, but no information is provided for how many articles were found for each topic area, were all papers found cited or only a portion of them?

Response: We initially reviewed relevant pre-clinical BDI research within the past 20 years. According to reviewers’ suggestions, we incorporated more in vitro studies on bile duct cell research and specifically, studies related to the use of biologically active factors in driving bile duct tissue regeneration. Thus, several more papers were cited. Moreover, categorizing our review as a systematic review was inaccurate, our review is written more like a literature review as we aimed to review the existing literature to identify the research gaps and propose future research directions of the field. Thus, we refrained from referring to our review as a systematic one and edited the indicated section to prevent confusion.

Reviewer 2 Report

This review paper summarizes the latest papers on artificial bile ducts and focuses on the issues of practical application. However, there are few references to summarize artificial bile ducts. Only the content of the latest references was similar to that of the review reported by Justin et al. in 2018. There was little new information about artificial bile ducts. The content of the Abstract dose not correspond to that of the text. There was no description in the text to review the content of the Abstract. As the context is not always related to the title or the section, the context should be largely reviewed. In addition, there was no concrete description to indicate active factors to promote tissue regeneration. Some description seems to be an idea of authors themselves, but it was not realistic. Taken together, I do not recommend this review paper to deserve this journal.

  1. Introduction

Lines 33-55: The authors mention the bile duct injury (BDI) as one case to be requested. How about the case of neoplasia of the biliary tract?

  1. Materials choice for artificial bile ducts

Lines 73-74: The authors mention that the previous references on bile duct supporting stents are excluded. However, there are some sentences of stents (lines: 110-112, 174, 176, 289, and 290), which confuses the readers.

Lines 76-77: The subjects "scaffold materials" and "cell seeded materials" are described in the text. But how about the subject "bioactive factor functionalization". It should be added.

Lines 100-101: In the previous references of artificial bile duct using decellularization technology, luminal organs, such as blood vessels and ureters, are used. Hassanein et al. reported that bile duct epithelial cells were identified in recellularization in the decellularized liver. Bile duct epithelial cells might exist in the liver-specific ECM. This point should be mentioned.

Lines 113-115: Why is the reference of Aikawa et al. introduced here?

Lines 122-125: The authors mention that the GelMA hydrogel provided an improved biocompatibility without detriment to the PCL mechanical properties. but what is the mechanism on the results of bone marrow-derived mesenchymal stem cell proliferation?

Line 129: Dose the phrase "biologically active" mean "bioactive factor functionalization" on the line 76? What "biologically active" is shown in the Li et al. reference?

Lines 131-134: Other problems of decellularization include the sources (for example, animal protection if made from animals) and quality control. The points should be described.

Line 136: The authors claim that degradable synthetic polymers are of poor biocompatibility. However, polymers are used in clinical practice. The reason why degradable synthetic polymers are of poor biocompatibility should be explained?

Lines 136-138: This was be not due to just a problem with degradable synthetic polymers, but must be the process of decellularization.

Line 139: What dose the wording “semi-synthetic materials” mean ? Tao et al. reported about the controlled release bFGF from the scaffold, while Liu et al. did about two-layered tubes incorporating cells. The concept is different to each other. Are they in the same category of semi-synthetic materials?

  1. Cell-seeded scaffolds and biocompatibility

Line 171: In culturing cell-seeded scaffolds, more appropriate references should be quoted to describe the factors that contribute to the bile duct regeneration.

Lines 176-177: Some methods to prepare primary bile duct cells from mice has been established (Yu Du et al., Hepatology, 2019). They should be quoted.

Lines 177-182: The authors’ statement way make readers misunderstand cells as using another cell as an alternative to bile duct epithelial cells. Mesenchymal stem cells (MSC) have been extensively used because they were important cells in the field of regenerative medicine.

Lines 189-196: I do not think these sentences to writing in the text.

Lines 196-197: Why did the authors introduce the cell transplantation?

Line 198: The word “in vitro” should be “in vivo”.

Lines 197-211: Whether or not the scaffold requires in cell seeding is very important issue in the regenerative medicine. In clinical practice, a scaffold without the cells seeding is also desired. The presence or absence of cell seeding depend on the objective. Miyazawa et al. have reported several references of successful bile duct regeneration on scaffolds without cell seeding. This is very important.

Lines 220-261: It is difficult to understand the relevance of these sentences to this section title.

Lines 260-261: It is difficult to understand the relevance of these sentences to this section title.

Lines 262-269: If the organoid can be used as a cell source, it should be added as one solution on the lines 176-177.

There is no description about the biocompatibility in the subtitle "Cell-seeded scaffolds and biocompatibility".

  1. Reasons for poor translatability and clinical inadequacies

What does the word “translatability” mean?

  • Limitations of scaffold materials

Lines 272-274: Some related references should be quoted.

Lines 295-297: Some related references should be quoted.

Lines 306-309: Does the phrase "above-mentioned necessary requirements" mean "Biocompatibility and biomechanical compliance and resilience" in lines 285-286? It might be difficult to resolve these problems with materials alone, but it will be improved with materials modification. The controlled release factors must be useful for cell- induced tissue regeneration. I think it risky to conclude that most materials cannot meet the necessary requirement. Biomaterials have a promising for standardized commercialization.

  • Limitation of cell-seeding techniques

   Lines 325-326: The sentences are repeated on the lines 244-248.

   Lines 341-343: Most of epithelial cells have fallen off from the gallbladder surgically removed due to ischemia of arterial dissection. A biopsy of the gallbladder is usually performed

when a malignant tumor is suspected. In any case, it would not be feasible to use the cells as cell sources.

Lines 332-353: I think that extrahepatic cholangiocyte organoids (ECO) might use in the basic research of the bile duct and will also in regenerative medicine in future. However, it is not related to this section. Limitations of cell-seeding techniques should be described, such as pre-transplant preparation, cell sources (cell quality assurance and tumorigenesis), immune response, etc.

Lines 354-368: This content must move to the third section (cell-seeded scaffolds and biocompatibility).

  1. Future research directions

Lines 373-374: Most BDIs were iatrogenic and mostly occurred at the extrahepatic bile duct. Oddi sphincter of vater's papillais preserved if a part of extrahepatic bile duct was removed and reconstructed.

Lines 386-389: Since pancreaticoduodenectomy performed for pancreatic cancer eliminates Vater's papilla, it would be ideal if the bile duct could be reconstructed including the Oddi sphincter. However, the Oddi sphincter is the smooth muscle and regulated by the parasympathetic nerves. I think that the regenerating neuromodulatory properties is very challenging.

Lines 392-411: It is not easy to simply link the development of bile duct to regeneration. The authors should consider the bile duct homeostasis or regeneration in bile duct disorders.

Lines 416-418: What does the phrase "The construction of artificial bile ducts based on the physiological mechanisms of bile duct formation" mean? It should be clearly described.

Lines 422-430: I think that this paper contains a unique perspective. However, there were no references to scientifically support the sentences. It is difficult for readers to imagine how to control the molecular mechanism.

Line 439: The word “bioactivity” is used in the text. However, it is unclear what the authors wanted to show about the "biologically active factors" described in the abstract.

Author Response

Response: We would like to take this opportunity to thank the reviewer for their expert opinion and in-depth critical assessment of our review article. The reviewer did not recommend our review for publication in its previous state, and we take this criticism seriously. Based on another review, the editors gave us a chance to substantially revise our manuscript, to address the comments raised and to lengthen the manuscript. We have carefully read and responded to each comment provided by the reviewer, and hope that the new version of the manuscript addresses the reviewer’s concerns.

Lines 33-55: The authors mention the bile duct injury (BDI) as one case to be requested. How about the case of neoplasia of the biliary tract?

Response: Cholangiocarcinoma needing tumor excision and removal of part of the bile duct could fall within the remit of bile duct replacement with tissue engineered artificial bile ducts. Intraductal papillary neoplasm is a possibility if it were detected early enough and was isolated to the bile duct and hadn’t spread to the entirety of the biliary tree. For the purpose of this review, we decided to focus on BDI as most pre-clinical studies utilize BDI models. Therefore, there is a limited amount of data available for bile ducts cancers. That being said, we did not discuss the possible applications of artificial engineered bile ducts in broader bile duct diseases, such as cancers, but agree that it is an important potential application. We have suggested neoplasia/carcinomas as potential applications for tissue engineered bile ducts, and have added it to the Future research directions section. Please see red text, lines 485-489.

Lines 73-74: The authors mention that the previous references on bile duct supporting stents are excluded. However, there are some sentences of stents (lines: 110-112, 174, 176, 289, and 290), which confuses the readers.

Response: Our original intention was to exclude research reporting physical stenting of the bile duct and now realize the confusion this caused. We have reworded the statement: “research was specifically related to tissue-engineered artificial bile ducts or bile duct scaffolds.”

Lines 76-77: The subjects "scaffold materials" and "cell seeded materials" are described in the text. But how about the subject "bioactive factor functionalization". It should be added.

Response: As part of expanding the manuscript, we have added a third section dedicated to the incorporation of biologically factors. We initially included the few studies on this topic within the other sections. However, after careful consideration, we decided to write a much larger third section, which highlights multiple avenues of research that have strong potential to be employed for tissue engineering of bile ducts. We refer the reviewer to sections 4 and 5.3 (limitations of biologically active factor incorporation) in the revised manuscript.

Lines 100-101: In the previous references of artificial bile duct using decellularization technology, luminal organs, such as blood vessels and ureters, are used. Hassanein et al. reported that bile duct epithelial cells were identified in recellularization in the decellularized liver. Bile duct epithelial cells might exist in the liver-specific ECM. This point should be mentioned.

Response: We thank the reviewer for providing this important comment. Indeed, we did not include decellularised luminal organs, and this does make up a proportion of the literature. We have selected the most recent examples to include within our manuscript, to keep it in theme with recent advancements. We also highlight common limitations of decellularized ECM scaffolds in tissue regeneration in this section. Please see red text, lines 120-128.

Lines 113-115: Why is the reference of Aikawa et al. introduced here?

Response: Our thanks to the reviewer for spotting this. It was a carry-over from early manuscript drafts that we missed removing during reference management. We have removed the sentence and reference from this section.

Lines 122-125: The authors mention that the GelMA hydrogel provided an improved biocompatibility without detriment to the PCL mechanical properties. but what is the mechanism on the results of bone marrow-derived mesenchymal stem cell proliferation?

Response: In the referenced work, the authors used biocompatibility to refer to GelMA’s ability to support seeded BMSC proliferation and survival. However, only a superficial assessment of cell growth was performed, no mechanistic experiments were presented. In multiple other studies, the gelatin RGD-motif contained within GelMA was shown to support of cell adhesion, proliferation, and spreading. In the study by Li et al., shortcomings of their design became apparent with the addition of the magnetic USPIO used for MRI imagine, but without assessment of GelMA-free materials as a control group, the exact benefit of GelMA on improving BMSC proliferation over the PCL scaffold was not further detailed. We have included an addendum to the revised manuscript to describe the proposed benefit of using GelMA has a cell substrate. Please see red text, lines 149-151.

Line 129: Dose the phrase "biologically active" mean "bioactive factor functionalization" on the line 76? What "biologically active" is shown in the Li et al. reference?

Response: Our use of ‘biologically active’ was for components of the scaffolds that possess inherent scaffold-cell or scaffold-protein interactions, thereby possessing biological activity – usually through the stimulation of intracellular pathways. We realized that the terms used could cause confusion and we have revised the terminology use throughout the manuscript to be more distinct in our meaning.

Lines 131-134: Other problems of decellularization include the sources (for example, animal protection if made from animals) and quality control. The points should be described.

Response: The problems of decellularization of ECM scaffolds have been included in the new text added to the relevant section. Please see lines 117-118. Additionally, zoonosis was an added limitation to the last entry (decellularized SIS) of the study summary table 1.

Line 136: The authors claim that degradable synthetic polymers are of poor biocompatibility. However, polymers are used in clinical practice. The reason why degradable synthetic polymers are of poor biocompatibility should be explained?

Response: We apologize for this oversight, it was unintentional to use ‘poor’ – as part of the revised manuscript, we have re-written section 2 on materials choices, which no longer includes the statement.

Lines 136-138: This was not due to just a problem with degradable synthetic polymers, but must be the process of decellularization.

Response: As part of the revised manuscript, we have re-written section 2 on materials choices, which has addressed the impact of decellularization on ECM-derived scaffolds.

Line 139: What dose the wording “semi-synthetic materials” mean ? Tao et al. reported about the controlled release bFGF from the scaffold, while Liu et al. did about two-layered tubes incorporating cells. The concept is different to each other. Are they in the same category of semi-synthetic materials?

Response: By semi-synthetic, we referred to synthetic polymer scaffolds that were modified to display/release biological compounds. As part of our revision, we have re-written section 2 on materials choices and have restricted the terminology to those more commonly utilized in the field – synthetic (degradable/non-degradable) and naturally-derived materials. Referrals to ‘semi-synthetics’ were adjusted to ‘functionalized synthetics.’

Line 171: In culturing cell-seeded scaffolds, more appropriate references should be quoted to describe the factors that contribute to the bile duct regeneration.

Response: According to the reviewers’ suggestions, we added a third section to the manuscript that describes biologically active factor incorporation (i.e., cytokines, hormones. and chemical compounds that convey beneficial cellular or tissue functionality), which discusses the factors that contribute to bile duct regeneration.

Lines 176-177: Some methods to prepare primary bile duct cells from mice has been established (Yu Du et al., Hepatology, 2019). They should be quoted.

Response: Please see the red text on line 263 in the revised manuscript for the reference to work by Du et al.

Lines 177-182: The authors’ statement way make readers misunderstand cells as using another cell as an alternative to bile duct epithelial cells. Mesenchymal stem cells (MSC) have been extensively used because they were important cells in the field of regenerative medicine.

Response: We carefully revised to manuscript to avoid further confusion, the Cell-seeded scaffolds has been divided into stem cell and stem cell-derived bile duct cell sections.

Lines 189-196: I do not think these sentences to writing in the text.

Response: The sentences were removed and/or rewritten as part of the section describing biologically active factor incorporation.

Lines 196-197: Why did the authors introduce the cell transplantation?

Response: The research team’s speculation of cell transplantation of BECs was introduced to suggest BECs as a potential cell source for cell-seeded scaffolds. We would like to re-iterate that this was a speculation made in the referenced article. We have now reworded the sentence to “The research team speculated that BECs differentiated in the above manner could be utilized to repair BDI” and removed mention of ‘injected.’

Line 198: The word “in vitro” should be “in vivo”.

Response: Our original statement was correct. The teams used in vitro data to justify their in vivo experiments. However, we have revised the sentence in case of further confusion to the reader.

Lines 197-211: Whether or not the scaffold requires in cell seeding is very important issue in the regenerative medicine. In clinical practice, a scaffold without the cells seeding is also desired. The presence or absence of cell seeding depend on the objective. Miyazawa et al. have reported several references of successful bile duct regeneration on scaffolds without cell seeding. This is very important.

Response: We agree. Our manuscript refers to the results by Miyazawa et al., and stated that they showed successful bile duct regeneration on scaffolds without cell seeding. Cell-free scaffolds include the use of functionalized materials, which we have dedicated a new section to in the revised manuscript.

Lines 220-261: It is difficult to understand the relevance of these sentences to this section title.

Response: The sentences were removed and/or rewritten as part of the section describing biologically active factor incorporation.

Lines 262-269: If the organoid can be used as a cell source, it should be added as one solution on the lines 176-177.

Response: This is the case as shown in early research using organoid-seeded collagen tubes and duct-structures. An explicit statement that suggests the use of organoids as potential cell sources has been added. Please see red text, lines 256-258.

There is no description about the biocompatibility in the subtitle "Cell-seeded scaffolds and biocompatibility".

Response: We originally intended to reflect the use of in vitro cell culture as a test of biocompatibility of scaffold materials. However, in the revised manuscript we described biocompatibility were relevant, and on a case-by-case basis. Therefore, the title was revised to ‘cell-seeded scaffolds.’

What does the word “translatability” mean?

Response: The section title was revised to ‘Current limitations and reasons for poor clinical translation.’

Lines 272-274; 295-297: Some related references should be quoted.

Response: Relevant references have been added.

Lines 306-309: Does the phrase "above-mentioned necessary requirements" mean "Biocompatibility and biomechanical compliance and resilience" in lines 285-286? It might be difficult to resolve these problems with materials alone, but it will be improved with materials modification. The controlled release factors must be useful for cell-induced tissue regeneration. I think it risky to conclude that most materials cannot meet the necessary requirement. Biomaterials have a promising for standardized and commercialization.

Response: Our original statement was overly negative, but that was not our intention. We concluded from the review of the literature that common issues with artificial bile duct materials inlcude incidence of leakage, stricturem or stenosis – but the current advancements have provided promising results that are helping to inform the field. The summary in-brief of the first section now reads as,

In summary, currently employed scaffold materials are promising in their application for the repair of BDI and have been imperative in informing the future development of scaffold materials but have been met by biomechanical and biocompatibility issues that may lead to biliary leakage, stricture, and stenosis. These major overarching limitations form the core difficulties in clinical translation of many tissue-engineered luminal tissues, including artificial bile ducts.”

Lines 325-326: The sentences are repeated on the lines 244-248.

Response: Our thanks for the reviewer indicating this error. The text was revised, and the repeated sentence removed.

Lines 341-343: Most of epithelial cells have fallen off from the gallbladder surgically removed due to ischemia of arterial dissection. A biopsy of the gallbladder is usually performed when a malignant tumor is suspected. In any case, it would not be feasible to use the cells as cell sources.

Response: We appreciate the reviewer’s expert insights and have added this as an important limitation. Please see lines 436-439.

Lines 332-353: I think that extrahepatic cholangiocyte organoids (ECO) might use in the basic research of the bile duct and will also in regenerative medicine in future. However, it is not related to this section. Limitations of cell-seeding techniques should be described, such as pre-transplant preparation, cell sources (cell quality assurance and tumorigenesis), immune response, etc.

Response: The inclusion of ECO demonstrated the current unknowns and limitations of using ECO-seeded scaffolds, particularly raising doubt over whether then remain as organoids after engraftment. Commonly debated limitations of cell-based therapies and cell-seeded scaffolds were introduced briefly at the beginning of the section, but we have chosen to focus on issues more specific to bile duct regeneration. We refer the reviewer to the revised text in section “5.2. Limitations of cell-seeding techniques.”

Lines 354-368: This content must move to the third section (cell-seeded scaffolds and biocompatibility).

Response: The sections were rewritten, and the text was moved.

Lines 373-374: Most BDIs were iatrogenic and mostly occurred at the extrahepatic bile duct. Oddi sphincter of vater's papillais preserved if a part of extrahepatic bile duct was removed and reconstructed.

Response: We thank the reviewer for this reminder. We have added this important distinction to the main text. Please see lines 488-491.

Lines 386-389: Since pancreaticoduodenectomy performed for pancreatic cancer eliminates Vater's papilla, it would be ideal if the bile duct could be reconstructed including the Oddi sphincter. However, the Oddi sphincter is the smooth muscle and regulated by the parasympathetic nerves. I think that the regenerating neuromodulatory properties is very challenging.

Response: Indeed, we fully agree that achieving full restoration of sympathetic nervous system control over tissue engineered Oddi sphincter and bile duct epithelium would pose a substantial challenge. A potential solution that may bridge the gap between transplant and re-innervation of the replaced bile duct sphincter is progress towards tissue engineered contractile smooth muscle layers, such as in research demonstrating esophageal muscle layer regeneration (Wang et al. 2018) and pyloric sphincter cell engraftment to restore functionality (Thurner et al. 2020). The challenge posed by nervous system control over bile duct function, the potential of smooth muscle cell restoration of sphincter function, and appropriate references have been added to the Future research directions section of the revised manuscript. Please see lines 508-519.

Lines 392-411: It is not easy to simply link the development of bile duct to regeneration. The authors should consider the bile duct homeostasis or regeneration in bile duct disorders.

Response: Our intention was to suggest that developmental processes form the inspiration for multiple tissue engineering techniques. Indeed, homeostasis and adult-tissue regenerative processes are equally as important and should also be prioritized considerations. We ensured appropriate referral within the rewritten section of the Future research directions sections.

Lines 416-418: What does the phrase "The construction of artificial bile ducts based on the physiological mechanisms of bile duct formation" mean? It should be clearly described.

Response: The original text was reworded, “The construction of artificial bile ducts based on materials that emulate or induce cell physiological mechanisms involved in bile duct homeostasis and regeneration have broad clinical application prospects.”

Lines 422-430: I think that this paper contains a unique perspective. However, there were no references to scientifically support the sentences. It is difficult for readers to imagine how to control the molecular mechanism.

Response: Our thanks for the reviewer’s comment on our perspective. We have added key references to the support the propositions made. Importantly, this includes the thorough review by Zohorsky and Mequanint (Tissue Eng Part B Rev. 2020) on the design of biomaterials to modulate Notch signaling and promote tissue engineering and tissue regeneration. Please see lines 545-546.

Line 439: The word “bioactivity” is used in the text. However, it is unclear what the authors wanted to show about the "biologically active factors" described in the abstract.

Response: According to the reviewers’ suggestions, we added a third section to the manuscript that describes biologically active factor incorporation (i.e., cytokines, hormones. and chemical compounds that convey beneficial cellular or tissue functionality). The text was revised to clarify the meaning of ‘bioactivity’ versus the incorporation of ‘biologically active factors.’

Round 2

Reviewer 1 Report

The authors have suitably addressed comments, it is recommended in future they better mark their revisions in manuscripts as they added entire new sections that were not indicated in red font.  The description of Figure 2 in the text is brief and could be improved.

Author Response

Reviewer 1:

The authors have suitably addressed comments, it is recommended in future they better mark their revisions in manuscripts as they added entire new sections that were not indicated in red font. The description of Figure 2 in the text is brief and could be improved.

Response: We thank the reviewer for their continued due diligence in helping to improve our manuscript. We have substantially expanded on the description of Figure 2 in the text, and have marked the section in red text (Lines 195-207).

Reviewer 2 Report

Dear. authors,

I make some suggestions to improve your text.

Line 141: The phrase "bile duct stents" should be "artificial bile duct”

Line 142: The phrase "stents" should be "artificial bile duct”

Line 143: The phrase "stents" should be "artificial bile duct”

Line 195: The word "stent" should be deleted

Line 198: The word " bile duct stents and" should be deleted.

Lines 214-219: Aikawa et al. reported that the artificial bile duct was implanted without cells. Zong et al. reported that the artificial bile duct was implanted with cells. The difference in results between with and without cells should be described clearly.

Lines 306-314: The sentence should be revised as follows. “Lin et al. induced differentiation from MSCs into BECs using TGF-β, FGF, EGF, and so on. Accordingly, the references should be quoted.

Lines 331-339: The sentence should be revised as follows. “Sampaziotis and colleagues induced differentiation from hiPSCs into CLCs using activin A, FGF2, BMP4 and so on.”

Line 370: The phrase "stents" should be "materials”

Line 371: The phrase " stent frame" should be "materials”

Author Response

Reviewer 2:

Dear. authors, I make some suggestions to improve your text. Line 141: The phrase "bile duct stents" should be "artificial bile duct”; Line 142: The phrase "stents" should be "artificial bile duct”; Line 143: The phrase "stents" should be "artificial bile duct”; Line 195: The word "stent" should be deleted; Line 198: The word " bile duct stents and" should be deleted.

Response: We thank the reviewer for their continued due diligence in helping to improve our manuscript. We have performed the above-mentioned adjustments to the revised manuscript (red text indicates additional wording where applicable).

Lines 214-219: Aikawa et al. reported that the artificial bile duct was implanted without cells. Zong et al. reported that the artificial bile duct was implanted with cells. The difference in results between with and without cells should be described clearly.

Response: The Aikawa reference was misplaced, our intention was to highlight work from the same group (Miyazawa et al.), which involved the comparisons of cell-free versus cell-laden scaffolds. The Aikwawa et al. reference was relocated to the appropriate section and further details were added regarding their cell-free polymer patch observations. (Please see the red text at lines 134-143).

For the expanded section on Zong et al, and Miyazawa et al., please see the red text at lines 219-241.

Lines 306-314: The sentence should be revised as follows. “Lin et al. induced differentiation from MSCs into BECs using TGF-β, FGF, EGF, and so on. Accordingly, the references should be quoted.

Response: We thank the reviewer for making suggestions to improve the clarity of our manuscript. The new text (please see red text at lines 319-322) reads as follows:

Lin et al. induced differentiation from BMSCs to BECs using cytokine regimen treatments, including transforming growth factor-β (TGF-β), fibroblast growth factor (FGF), and epidermal growth factor (EGF) supplementation, and so on. The resulting BECs expressed specific markers such as cytokeratin-7, cytokeratin-19, and glutamyl transferase [39].

Lines 331-339: The sentence should be revised as follows. “Sampaziotis and colleagues induced differentiation from hiPSCs into CLCs using activin A, FGF2, BMP4 and so on.”

Response: We once again thank the reviewer for their suggestions. The new text (please see red text at lines 337-339) reads as follows:

Sampaziotis and colleagues induced differentiation from human iPSCs to CLCs using a "five-step method" that involved the addition of activin A, FGF-2, bone morphogenetic protein-4 (BMP4), and so on.

Line 370: The phrase "stents" should be "materials”; Line 371: The phrase " stent frame" should be "materials”

Response: We have performed the above-mentioned adjustments to the revised manuscript (red text indicates change of wording, line 379).